# Splicing Characterization and Isoform Switch Events in Human Keratinocytes Carrying Oncogenes from High-Risk HPV-16 and Low-Risk HPV-84

**DOI:** 10.3390/ijms24098347

**Published:** 2023-05-06

**Authors:** Maryam Nasiri-Aghdam, Mariel Garcia-Chagollan, Ana Laura Pereira-Suarez, Adriana Aguilar-Lemarroy, Luis Felipe Jave-Suarez

**Affiliations:** 1División de Inmunología, Centro de Investigación Biomédica de Occidente, Instituto Mexicano del Seguro Social, Guadalajara 44340, Mexico; maryamnasiri.bio@gmail.com (M.N.-A.); adry.aguilar.lemarroy@gmail.com (A.A.-L.); 2Departamento de Biología Molecular y Genómica, Centro Universitario de Ciencias de la Salud, Universidad de Guadalajara, Guadalajara 44340, Mexico; 3Instituto de Investigación en Ciencias Biomédicas, Centro Universitario de Ciencias de la Salud, Universidad de Guadalajara, Guadalajara 44340, Mexico; maye_999@hotmail.com; 4Department of Microbiology and Pathology, Centro Universitario de Ciencias de la Salud, Universidad de Guadalajara, Guadalajara 44340, Mexico; ana.pereira@academicos.udg.mx

**Keywords:** high-risk HPV, low-risk HPV, mRNA processing, splicing, cervical cancer

## Abstract

Infection of epithelial cells with high-risk HPV (HR-HPV) types, followed by expression of virus oncogenic proteins (E5, E6, and E7), leads to genomic imbalance, suppression of tumor inhibitors, and induction of oncogenes. Low-risk HPV (LR-HPV) may slow the rate at which cervical cancer spreads to an invasive stage since co-infection with LR-HPV is linked to a decreased risk of future invasive cancer than infection with HR-HPV alone. We then propose that cancer-progressing changes may be distinguished through identifying the functional differences between LR-HPV and HR-HPV. Lentiviral strategies were followed to establish HaCaT cells with constitutive expression of HPV oncogenes. RNAseq experiments were designed to analyze the transcriptome modulations caused by each of the E5, E6, and E7 oncogenes of HPV-16 and HPV-84 in HaCaT cells. We identified enhanced RNA degradation, spliceosome, and RNA polymerase pathways related to mRNA processing. ATTS (alternative transcription termination site) was discovered to be more prevalent in cells with HPV-16E5 than HPV-84E5. In HPV-16E6-infected cells, ATTS gain was significantly higher than ATTS loss. Cells with HPV-16E7 had more isoforms with intron retention (IR) than those with HPV-84E7. We identified switches in ADAM10, CLSPN, and RNPS1 that led to greater expression of the coding isoforms in HR-HPV. The results of this work highlight differences between LR-HPV and HR-HPV in mRNA processing. Moreover, crucial cervical cancer-related switch events were detected.

## 1. Introduction

Viral infection accounts for approximately 12–20% of all cancers [1,2]. Human papillomavirus (HPV) is one of the most common cancer-causing viruses and a significant etiological factor for many malignancies, including cervical cancer, anal cancer, vulvar and vaginal cancer, penile cancer, head and neck cancer, and skin cancer [3]. Currently, 229 different HPV types are identified and categorized as high-, low-, or unknown-risk, depending on their carcinogenic potential [4,5]. Infection of epithelial cells with high-risk HPV types (HR-HPV), followed by the expression of viral oncogenic proteins (E5, E6, and E7), leads to genomic imbalance, suppression of tumor inhibitors, and induction of oncogenes. This process eventually drives neoplastic progression, which can take years to conclude [6]. The HPV vaccine may not protect against all HR-HPV strains and may not benefit women who already have the virus [7]. Although low-risk HPV (LR-HPV) proteins can alter cell proliferation, apoptosis, and immortalization, their carcinogenic potential is lower than that of HR-HPV proteins [8]. LR-HPV infections cause benign cervical lesions, such as common genital warts, that rarely progress towards cancer [9]. LR-HPVs may slow the rate at which cervical cancer spreads to an invasive stage since co-infection with LR-HPVs decreases the risk of future invasive cancer compared to infections with HR-HPVs alone [10]. Thus, it is proposed that through identifying the functional differences between LR-HPVs and HR-HPVs, it may be possible to distinguish cancer-progressing alterations.

A crucial stage in the post-transcriptional control of gene expression is alternative splicing (AS) of mRNAs. This mechanism significantly modifies cell function and proteome diversity through regulating the expression of various isoforms at particular times and in specific cell types. In this sense, through altering the mRNA processing machinery, cancer cells can gain advantages [11]. Therefore, mRNA processing regulators are a new class of oncoproteins or tumor suppressors because they can control disease progression through altering the RNA isoforms associated with the primary cancer processes [12]. Cancer cells and each cancer type and subtype have unique splicing patterns, making AS a new hallmark of cancer [11,13]. With the advance of bioinformatic tools, it is now possible to research eight different AS mechanisms, including alternative transcription start sites (ATSS), alternative transcription termination sites (ATTS), alternative three-end acceptor sites (A3), alternative five-end donor sites (A5), exon skipping (ES), intron retention (IR), mutually exclusive exon (MEE), and multiple exon skipping (MES) [14]. Moreover, detecting isoform switch events and their predicted consequences for individual genes may help clarify splicing dysregulation during HPV infection.

Although there has been some research on the alternative splicing processes by HR-HPV types, modulation of AS through the proteins encoded with LR-HPVs receives far less attention. Splicing events generated by E5, E6, and E7 oncoproteins of HPV-84 and HPV-16 were examined for the first time to highlight the differences between HR- and LR-HPVs; these modulations were described in eight distinct splicing mechanisms and individual isoform switch events.

## 2. Results

### 2.1. Analysis of KEGG Pathways Showed High Enrichment of Pathways Involved in the mRNA Processing

To determine potential KEGG pathways and gene sets linked with HR- and LR-HPV types, the expression data of the HaCaT transduced cells with the oncogenes of HPV-16 and HPV-84 were subjected to GSEA analysis. Over-represented pathways with normalized gene set enrichment scores (NES) ≤ 1 are illustrated in Figure 1. Among them, mRNA processing pathways, including spliceosome, RNA degradation, and RNA polymerase, were repeatedly enriched between groups. Interestingly, HaCaT-16E5 cells have overexpression of genes involved with RNA degradation, whereas HaCaT-84E5 does not exhibited this pathway as an enriched one. In HaCaT cells transduced with HPV-16 and -84 oncogenes, enrichment of genes associated with spliceosome was observed when the E6 and E7 oncogenes were introduced. In addition, E6 and E7 from HPV-84 induced the expression of RNA polymerase genes, in contrast to E6/E7 from HPV-16, which induced the expression of genes related to the RNA degradation pathway. The gene set enrichment analysis results highlight the importance of mRNA processing pathways; therefore, these biological processes were further investigated as outlined in the following section.

### 2.2. Splicing Mechanisms Demonstrate Subtle Differences throughout the Genome

Genome-wide alternative splicing was investigated in the eight previously identified categories, and the results make it quite evident that not all alternative splicing events were equally used (Figure 2). In all comparisons, ATSS was the most prominent event, while MEE was the least frequent, occurring only in cells with the E5 and E7 oncogenes of HPV-84. In the cells carrying E5 of HPV-16, there were more significant ATTS gains than losses, with a difference of about five significant isoforms. However, the losses outweighed the gains in the A5. The most notable HPV-84 event was more significant ATSS loss than gain, which for HPV-16 was not seen. When comparing HPV-16 and HPV-84, HPV-84 had a greater loss in MES but fewer significant isoforms with ATTS, ES, and IR events than HPV-16. The most noticeable variation seen in cells with E6 was fewer isoforms with ATTS losses in HPV-16 than in HPV-84. Finally, the cells containing E7 from HPV-16 exhibited a higher number of isoforms with IR gains than those with the E7 from HPV-84, which shows a higher number of isoforms with MES losses.

In Figure 3, the uneven utilization within each comparison is shown through calculating the percentage of events that are *gains* for each AS type (as opposed to *loss*). It is evident from this plot that the overall trend in the usage of alternative splicing is the same in the two HPV types. However, there are some variances. We can, indeed, see that ATTS has a much more significant skew in HPV-16E6 compared to HPV-84E6.

### 2.3. Overall Predicted Switch Consequences Are Different

According to coding potential, identified domains, intron retention, sensitivity to non-sense mediated decay (NMD), and ORF length, isoform switch effects were predicted and characterized (Figure 4). The cells with HPV-16E5 oncogene exhibit increased isoforms with domain loss and sensitivity to NMD. Moreover, compared to HPV-84E5, HPV-16E5 showed a higher number of isoforms with shorter or longer ORF. HPV-16E6 induced more coding transcripts than HPV-84, where most switches are non-coding. One isoform with a domain switch was observed for HPV-16E6. Additionally, the number of isoforms with complete ORF gain and those with longer ORF are approximately higher in HPV-16E6 compared to HPV-84E6. Although the overall pattern in cells expressing E7 is identical for HPV-16 and HPV-84, HPV-84 has a more significant proportion of isoforms with domain loss, whereas HPV-16 has a higher proportion of NMD-insensitive isoforms.

### 2.4. Switch Overlaps and Increased Usage of Coding Isoforms Are Detected

The general overlaps in isoform switches for all three oncogenes from HPV-16 and HPV-84 are illustrated in Figure 5A–C. There was little overlap across all comparisons, indicating that each HPV type’s switch events are distinct. A minor overlap was seen between HPV-16E5 and HPV-84E5 for only one gene: ABHD14B (Figure 5D). The coding isoform ENST00000395008 of this gene had increased usage in the control group, and neither HPV-16E5 nor HPV-84E5 expressed this or any other coding isoforms of this gene. However, the overlap between HPV-16 and HPV-84 for E6 and E7 oncogenes approached three genes. Both HPV-16 and HPV-84, which are E6-transformed cells, had an elevated expression of at least one of the coding isoforms of AARS2, BZW2, and NUP205. A novel coding isoform of PCIF1 was upregulated in the control group versus HPV-16E6 and HPV-84E6. The DLST gene was the only gene where an isoform switch increased the isoform expression in both HPV-16E7 and HPV-84E7. When comparing the control group to HPV-16E7 and HPV-84E7, elevated expression of the other three genes—ABHD14B, RPUSD4, and TRAF7—was observed. Figure 5E listed all coding isoform switches when comparing HPV-16 to HPV-84 for all three oncogenes.

### 2.5. Isoform Switch Events in Specific Genes Are Notable

We discovered three events in the genes ADAM10, RNPS1, and CLSPN that were previously reported to play a role in the progression or prognosis of cervical cancer. The switch plots of these three genes are illustrated in Figure 6A–C. The first switch event occurred in ADAM10, where HPV-16E5 expressed the coding isoform ENST000000260408 more strongly than HPV-84E5. When comparing HPV-16E6 to HPV-84E6, two events were detected. The isoform ENST000000318121 was observed in CLSPN to be more abundant in HPV-16E6. In the third event, RNPS1, the expression of ENST000000566397 was higher in HPV-16E7.

## 3. Discussion

The maturation, processing, and alternative splicing of mRNAs are necessary for regulating gene expression and preserving proteome diversity. The development of cervical cancer is thought to be significantly influenced via HPV-mediated alternative splicing, and these alterations can be utilized as diagnostic biomarkers and targets for therapeutic strategies [15]. On the other hand, putative disease-causing splice isoforms might provide crucial details about cancer development and help develop therapeutic approaches [16]. Due to the importance of such alterations, we hypothesized that high- and low-risk HPV types might differ in splicing mechanisms and isoform switch events, enabling us to distinguish between non-cancerous and actual cancer driver events. We constructed six keratinocyte cellular models, one for each of the E5, E6, and E7 oncogenes derived from high-risk HPV-16 and low-risk HPV-84. The expression data of these models were initially used to perform gene set enrichment analysis, followed by extensive alternative splicing analysis and detection of individual and generic patterns of isoform switch events. Firstly, key pathways were identified via gene set enrichment analysis, with all being strongly related to mRNA processing and transcription regulation. Among these are RNA degradation, spliceosomes, and RNA polymerase. Except for cells containing HPV-84 E5, they were found to be enriched in all experimental groups. Several key splicing factors were found with altered expression in cervical cancer, primarily from SRSF and hnRNP protein families, which were reviewed in detail in several recently published articles [17,18,19].

RNA degradation aids ribonucleotide recycling, while also performing surveillance through eliminating aberrant RNA that might produce damaging proteins [20]. There are scarce studies on modifications of RNA degradation-related genes in cervical cancer. A 2020 study showed that deleting EDC4 (Enhancer of mRNA decapping protein 4) in cervical cancer cells enhanced sensitivity to cisplatin and inhibited cell proliferation produced via cisplatin and DNA damage [21]. This finding could imply that EDC4 is a new target to avoid chemotherapy resistance in cervical cancer. In a study on N6-methyl-adenosine (m6A) mediated via METTL3 (methyl-transferase-like 3), it was discovered that METTL3 promotes the stability of HK2 (hexokinase 2) through m6A modification, thereby promoting the Warburg effect (also known as aerobic glycolysis), which could lead to a new insight for the treatment of cervical cancer [22]. Concerning the RNA polymerase pathway, it was demonstrated that Pol III dysregulation stemming from oncogenes or tumor suppressors is present in many malignancies and contributes to carcinogenesis [23]. In summary, studies on mRNA processing pathways, specifically on RNA degradation and RNA polymerase in cervical cancer, are limited. Further research is needed to identify their function in the progression and treatment of cervical cancer.

The enrichment of mRNA processing pathways piqued our interest in further investigating mRNA processing activities in our cell models. Following the pipeline defined using IsoformSwitchAnalyzeR, we identified eight alternative splicing events in a number of significant isoforms. Notably, ATTS is more common in cells carrying HPV-16E5 than in cells harboring HPV-84E5. ATTS, on the other hand, is demonstrated to have the most notable alteration in cells with E6, with fewer isoforms with ATTS being detected in HPV-16 compared to HPV-84. Moreover, in HPV-16E6, ATTS gain is significantly higher than ATTS loss. Generally, using ATTS produces transcripts with different 3′ ends or even transcripts with different coding regions [24]. However, the stability of the transcripts generated from ATTS can vary depending on the transcript and the cellular context [25]. Alternative transcription start and termination sites were previously identified as the primary drivers of transcript isoform diversity across human tissues [26]. According to Kim et al., the distribution of ASE types differed between malignant and normal tissues, with cancer cells showing less exon skipping but more alternative start or end sites than normal cells [27]. ATTS dysregulation is a common occurrence in cancer, and it can result in increased oncogene expression and decreased tumor suppressor gene expression [28]. Cervical cancer was found to have dysregulation in two ATTS regulatory factors: cold-inducible RNA binding protein (CIRP, also known as CIRBP or A18 hnRNP), which binds and stabilizes pro-survival gene transcripts, and RBBP6, which suppresses polyadenylation [29]. Overall, ATTS seems to play an important role in cervical cancer, even distinguishing between high- and low-risk HPV types; however, further study is required to back up this finding.

A further finding in our study concerns E7-containing cells where HPV-16 has a more significant number of isoforms with IR than HPV-84. IR appears to be necessary at several phases of cell differentiation and development. Intron-retaining transcripts can be detained in the nucleus (a process known as intron detention) and destroyed via nuclear degradation mechanisms. As introns often include in-frame premature termination codons, when preserved in the mature mRNA transcript and exported to the cytoplasm, they can be identified via the cytoplasmic surveillance machinery and eliminated via nonsense-mediated decay (NMD). Furthermore, the intron retention mechanism may yield non-coding RNA(s) implicated in controlling oncogenes and tumor suppressor genes. [30]. Except for breast cancer, it has been reported that retention of alternative introns is enriched in 16 different cancer types; however, cervical cancer was not included in this study [31]. In our investigation, IR events distinguished between high- and low-risk E7; nonetheless, its role in cancer progression requires additional research.

In our study, we found a more significant number of isoforms with domain loss, sensitivity to NMD, and shorter or longer ORFs, primarily mediated by high-risk HPV16 oncogenes. It is now well understood that viral proteins influence the cellular splicing machinery, producing RNA isoforms with carcinogenic properties [17,18]. Furthermore, alternative splicing contributes to virally induced cancer progression through increasing the expression of proteins implicated in proliferation and immune response [32]. We focused on three isoform switch events in ADAM10, CLSPN, and RNPS1 that led to greater expression of the coding isoforms in high-risk HPV.

These genes have previously been linked to cervical cancer. The proteolytic cleavage of NKG2D ligands (NKG2DL) on the cancer cell surface via ADAM10 can impair the recognition of cancer cells by T or NK cells [33]. In cervical cancer, ADAM10 is the target of miR-140-5p, which is controlled by the SNHG20 lncRNA. The inhibition of SNHG20 can reduce ADAM10 protein expression, resulting in decreased cervical cancer cell proliferation [34]. The second gene—CLSPN—regulates the ATR/Chk1 signaling axis in the G2 DNA damage checkpoint [35]. In histological and cytological samples, claspin expression is substantially associated with HR-HPV infection and lesion grade, which could be therapeutically helpful in detecting HPV-related cervical lesions [36]. The third isoform switch occurred in RNPS1, which is an essential regulator of the splicing process previously found to be overexpressed in cervical cancer cells. Alternative splicing controlled by RNPS1 favors an active Rac1b/RhoA signaling axis, possibly contributing to cervical cancer cell invasion and metastasis [37]. Given the relevance of isoform switches in the control of coding and non-coding isoforms, it seems necessary to consider these switch events in future transcriptome analyses.

## 4. Materials and Methods

### 4.1. Obtaining Keratinocytes Carrying the E5, E6, and E7 Oncogenes of HPV-16 and HPV-84

Briefly, E5, E6, and E7 were amplified using specific primers from genomic DNA extracted from cervical biopsies of women infected with HPV16 or 84. The amplified ORFs were subcloned into the lentiviral expression vector pLVX-Puro and verified via sequencing. Lentiviral particles were produced using the acquired plasmids pLVX-16E5, pLVX-16E6, pLVX-16E7, pLVX-84E5, pLVX-84E6, and pLVX-84E7. In the following step, HaCaT cells were individually infected with the lentiviral particles containing each viral gene. Transduced cells were selected through puromycin addition for three weeks. Transduced HaCaT cells were named HAC16E5, HAC16E6, HAC16E7, HAC84E5, HAC84E6, HAC84E7, and HACPLVX [38]. HaCaT cells were initially kept frozen in liquid nitrogen until use. When the cultures were prepared, the cells were thawed at 37 °C, centrifuged at 1000× *g* rpm for 10min to remove excess DMSO (dimethyl sulfoxide), suspended in fresh medium, and cultured in 75 cm2 culture flasks. Cell lines were grown in Invitrogen’s GIBCO™ DMEM (Dulbecco’s Modified Eagle Medium) medium with L-glutamine (584 mg/L), sodium pyruvate (110 mg/L), and D-glucose (4.5 g/L). The medium was supplemented with penicillin (100 U/mL), streptomycin (100 μg/mL), and 10% fetal bovine serum. The cultures were maintained at a confluence of 70–80%; upon reaching this level, the cells were detached with 0.25% trypsin (EDTA 380 mg/L) and counted to make the necessary dilutions for the experiments. Culture passages were created according to the cell duplication rate. Cell cultures were maintained in a humidified incubator at 37 °C in a 5% CO_2_ atmosphere.

### 4.2. RNA Extraction and Sequencing

Total RNA was obtained from 5 × 10^6^ cells of each cell line in three replicates using the PureLinkTM RNA Mini Kit extraction system (Ambion, Naugatuck, CT, USA). Cells were washed two times with PBS, and cell lysis was performed. The lysate obtained was column purified, and the RNA was eluted in water. The RNA concentration and integrity were evaluated with the Agilent 2100 bioanalyzer using RNA nano chips. RT-qPCR confirmed the expression of E5, E6, and E7 through specific primers. Sequencing of total mRNA was performed using the Illumina Nova-Seq 6000 platform with a 150 bp paired-end method, where most reads have a Phred quality score greater than 35.

### 4.3. Differential Expression and Gene Set Enrichment

Data quality was verified using the FastQC v.0.11.9 tool [39] on the Galaxy platform available at https://usegalaxy.org/, accessed on 5 April 2021 [40]. Reads were mapped to the hg38v38 human genome using STAR v.2.7.9a [18] tool. Mapped reads were filtered based on the mapping quality utilizing the Qualimap v.2.2.1 tool [41]. The overall mapping rate was about 92%. The STAR-aligned reads were used separately as DESeq2 v.1.36.0 [42] input files for differential expression calculation between our experimental groups. After data pre-processing, the normalized counts from DESeq2 were uploaded to the GSEA v. 4.3.2 desktop application (https://www.gsea-msigdb.org/, accessed on 21 December 2021 [43,44], and enrichment analyses were performed for the Kyoto Encyclopedia of Genes and Genomes (KEGG) pathways [45]. The number of permutations was set to 1000, and the permutation type was selected as “gene_set”. Pathways with a normalized enrichment score >1 or <−1 at FDR < 10% were considered significant.

### 4.4. Transcript Assembly

For accurate reconstruction of all expressed isoforms of each gene, and to estimate the relative abundance of those isoforms, the StringTie v2.2.0 transcript assembler was used [46]. For this aim, the STAR alignment file of all reads from each sample in BAM format was taken as the input for StringTie to assemble transcripts. All the genes found in any samples were merged to create a single assembly. After merging, a second run of StringTie was performed to recalculate the abundance of the merged transcripts.

### 4.5. Isoform Switch Events and Alternative Splicing Analysis

IsoformSwitchAnalyzeR was used to detect and visualize the mechanisms of alternative splicing and individual isoform switch events [14]. In the pipeline, the transcript assembly files for each sample, the merged-GTF file from StringTie, and the reference GTF file (Homo_sapiens.GRCh38.102.gtf) were used to construct the isoformSwitchAnalysisPart1 object [47]. For the filtering step, a gene expression cutoff of 3 and an isoform expression cutoff of 1 were applied. The Isoform switch test was implemented using DEXSeq via IsoformSwitchAnalyzeR [48]. To predict the consequence of each isoform switch event in coding, non-coding, and nonsense-mediated decay (NMD) sensitive categories, data from external annotation sources, including Pfam for protein domains [49], SignalP for signal peptides [50], CPAT for potential coding sequences [51], and IUPred2A for prediction of intrinsically disordered regions [52], were then imported and integrated to the IsoformSwitchAnalyzeR object. To measure the effect size in isoform usage, IsoformSwitchAnalyzeR uses isoform fraction (IF) values, which quantify the fraction of the parent gene expression originating from a specific isoform (calculated as isoform_exp/gene_exp). Finally, the isoform(s) used frequently (positive dIF) is compared to the isoform(s) used infrequently (negative dIF) to identify potential functional consequences of the isoform switch. Only switch events in the coding isoform with a significant IF were considered when filtering out the isoform switch events. Additionally, all switch events were verified twice in the ENSEMBL database, and those in coding isoforms that translated to the same number of amino acids were eliminated. The events that did not affect the protein domains, signal peptides, or intrinsically disordered areas were also removed from the final table of coding isoforms. The following steps involved performing an alternative splicing analysis for the following eight categories: alternative 3′ acceptor sites (A3), alternative 5′ splice sites (A5), alternative transcription start sites (ATSS), alternative transcription termination sites (ATTS), exon skipping (ES), intron retention (IR), mutually exclusive exons (MEE), and multi-exon skipping (MES), both at genome-wide and isoform levels.

## 5. Conclusions

Our findings point to key differences produced by the oncogenes E5, E6, and E7 of HPV-16 and HPV-84. We focused primarily on mRNA processing pathways, where we discovered specific enhanced splicing processes and isoform switch events in HPV-16 that may favor cancer progression. These changes, in addition to providing information on biological processes implicated in cervical cancer pathways, may be beneficial for identifying prospective biomarkers and therapeutic targets. Nonetheless, only large prospective studies may determine the actual clinical and diagnostic utility of these alterations in cervical cancer.

## Figures and Tables

**Figure 1 ijms-24-08347-f001:**
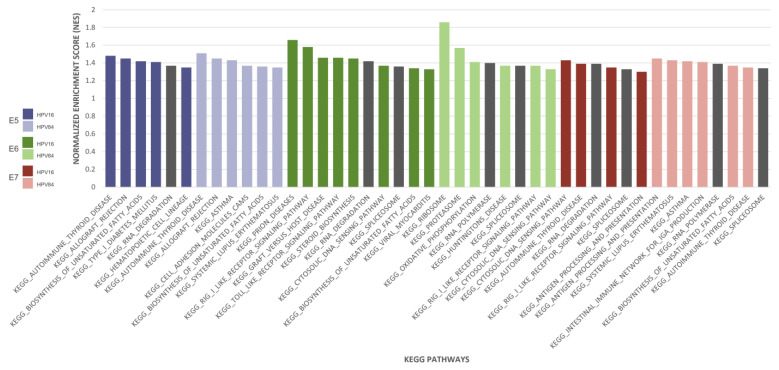
Gene set enrichment analysis (GSEA). Plot shows normalized enrichment scores of top KEGG pathways. Pathways related to mRNA processing are highlighted in dark grey.

**Figure 2 ijms-24-08347-f002:**
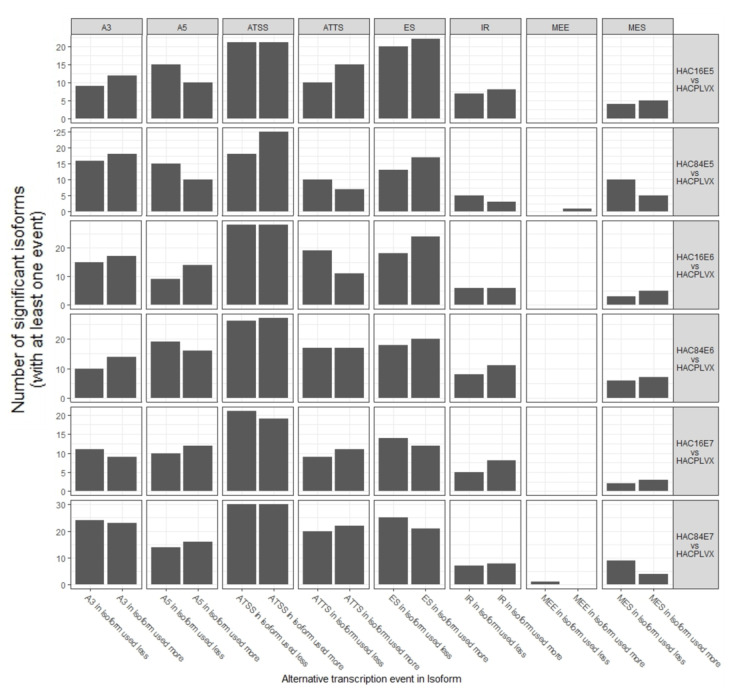
Alternative transcription events in isoforms. Plot shows eight AS mechanisms, including alternative transcription start sites (ATSS), alternative transcription termination sites (ATTS), alternative 3-end acceptor site (A3), alternative 5-end donor site (A5), exon skipping (ES), intron retention (IR), mutually exclusive exon (MEE), and multiple exon skipping (MES). HAC: HaCaT cells.

**Figure 3 ijms-24-08347-f003:**
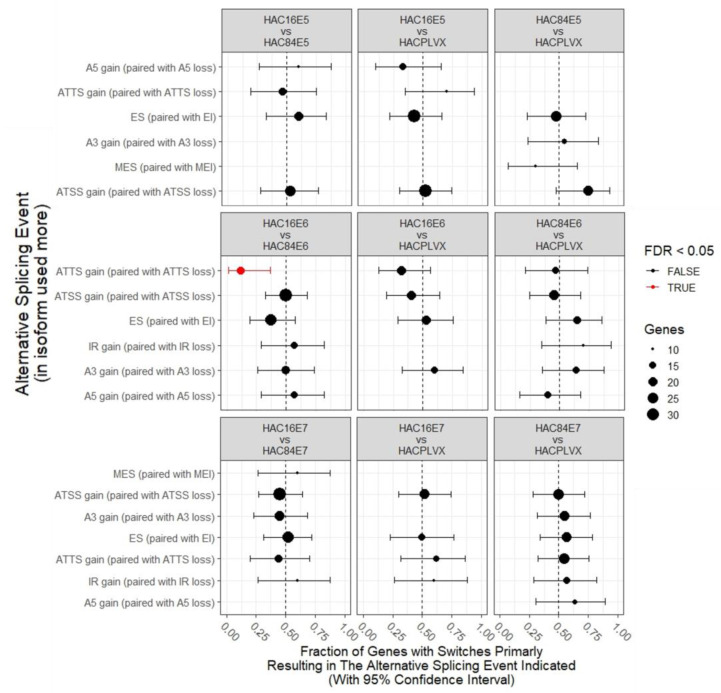
Gains or losses in alternative splicing events.

**Figure 4 ijms-24-08347-f004:**
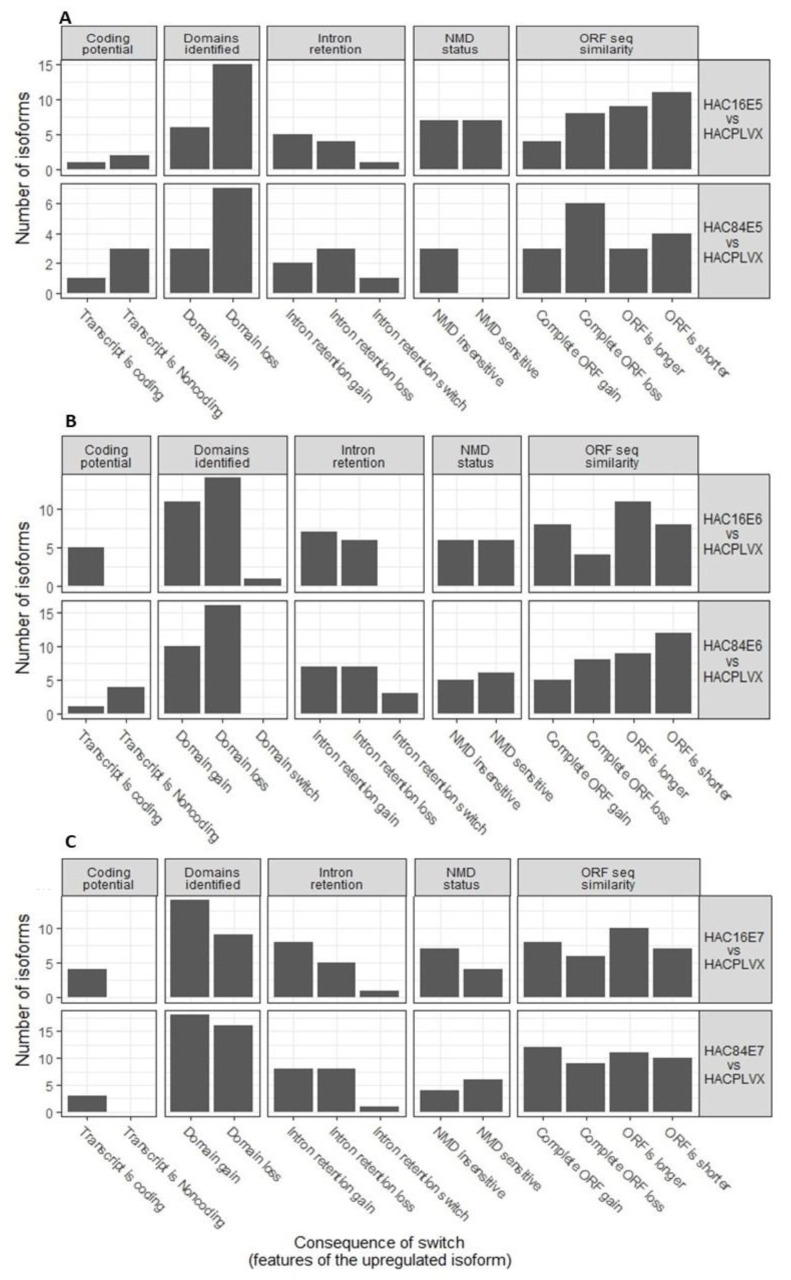
Overall isoform switch consequences. (**A**) for E5 from HPV-16 and HPV-84 versus control. (**B**) for E6 from HPV-16 and HPV-84 versus control. (**C**) for E7 from HPV-16 and HPV-84 versus control.

**Figure 5 ijms-24-08347-f005:**
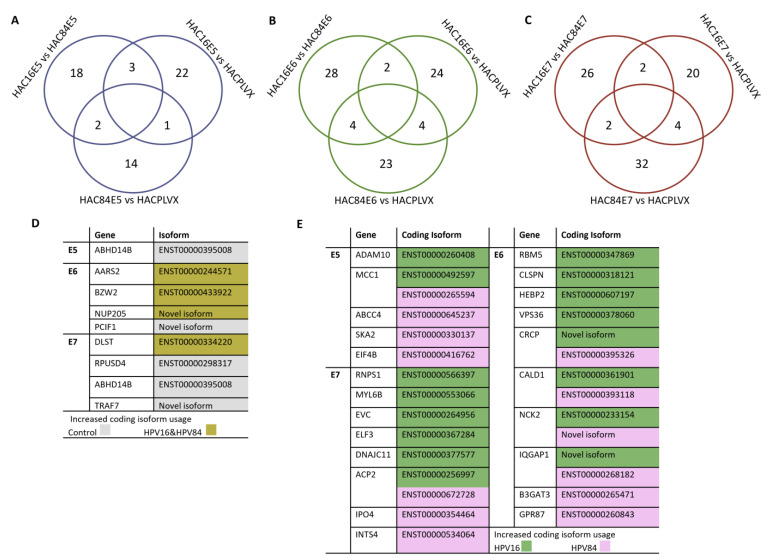
Isoform switch events reveal a little overlap between isoforms induced by each HPV oncogene. Overlaps between HPV16 and HPV84 are separated based on oncogene type (**A**) for E5, (**B**) for E6, and (**C**) for E7. List of genes with increased coding isoform usage (**D**) control versus HPV16 and HPV84. (**E**) HPV16 versus HPV84.

**Figure 6 ijms-24-08347-f006:**
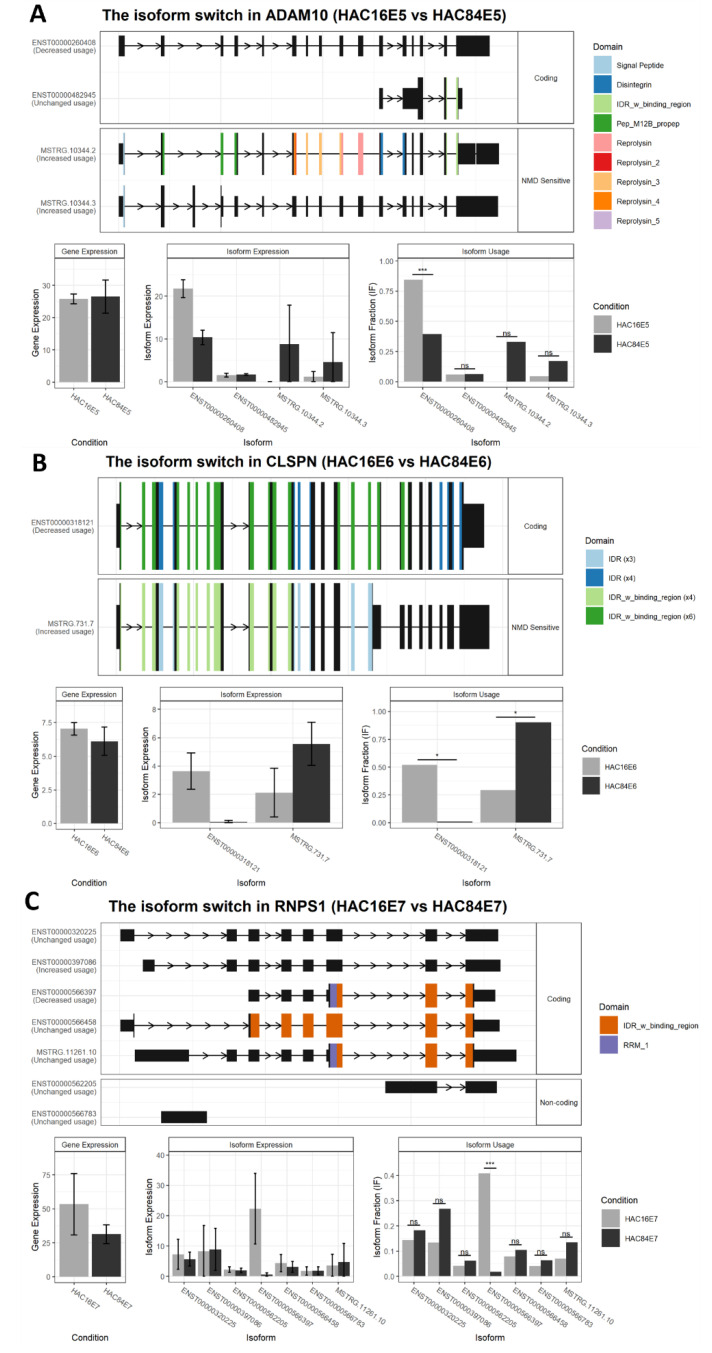
Isoform switch plots for genes previously related to cervical cancer when comparing HPV-16 to HPV-84: (**A**) ADAM10; (**B**) CLSPN; (**C**) RNPS1; *—*p*-value < 0.05, ***—*p*-value <0.001, ns—no significant difference.

## Data Availability

All raw and processed data for this project was deposited in the Gene Expression Omnibus (https://www.ncbi.nlm.nih.gov/geo, accessed on 30 August 2022) under accession number GSE228187. Scripts for differential expression and splicing analysis were released in the following GitHub repository: https://github.com/MaryamNasiriAghdam/MNasiri_HPV, accessed on 14 February 2022.

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
