# Peer review of "Splicing Characterization and Isoform Switch Events in Human Keratinocytes Carrying Oncogenes from High-Risk HPV-16 and Low-Risk HPV-84"

_ijms, 2023, doi:10.3390/ijms24098347_

Round 1

Reviewer 1 Report

This is a very nice study.

HPV-16 is a well-studied model for alternative RNA processing. Not aware that these events have been previously described in HPV-84.

Please spell check the final version.

Author Response

RESPONSE TO REVIEWERS' COMMENTS

Thank you very much for all the comments made, they really allowed us to make a better version of the manuscript. The changes suggested by you were done, as described below:

Reviewer 1 Comments

Point 1: HPV-16 is a well-studied model for alternative RNA processing. Not aware that these events have been previously described in HPV-84.

Response 1: As you mentioned, other studies have previously evaluated the alterations in the RNA-processing mechanism triggered by HPV-16. However, to our knowledge, this study is the first to compare mRNA processing differences between high-risk and low-risk HPV. We think those differences could be relevant to cancer development.

Point 2: Please spell check the final version.

Response 2: We have asked a native English speaker to final spell check the paper.

Thank you for your time and consideration.

Luis Felipe Jave Suárez, Maryam Nasiri Aghdam,

Reviewer 2 Report

The manuscript entitled “Splicing characterization and isoform switch events in human keratinocytes carrying oncogenes from high-risk HPV-16 and low-risk HPV-84” by Nasiri-Aghdam et al. is of potential interest to those in the field of HPV genetics.  While the English in the manuscript is good (improvements are needed to correct typos and grammatical errors), I am unable to evaluate the merits of this manuscript as critical information is either omitted or not explained well.  Therefore, I am going to refrain from commenting on minor points and focus on substantial comments as this manuscript needs to be revised before consideration.

1.  I did not find any discussion or mention of the sample size in their research.  Did they obtain total RNA from one biological replicate of 5 x 106 cells for each of their transfected cells?  How many datasets were being analyzed?

2.  There is very little explanation in the methods about how they did the KEGG analysis, the parameters chosen, and what FDR <10% means. This may be appropriate for those in the field, but I feel a bit more explanation on how the analysis was performed and why they chose the genes/pathways to represent in Figure 1.  While the authors had a focus for RNA processing pathways, Figure 1 shows even higher NES scores for “Ribosome” and various autoimmune conditions.  Why are these illustrated and/or what do they mean?  There’s no discussion of these other data in Figure 1.

3.  In the results for Figure 2, alternative splicing mechanisms are shown and the discussion mentions “gains and losses”.  Compared to what?  Each other?  Is the point that the HPV16 strains use more or less AS strategies than the HPV84 strains?  What does this mean (lines 105-107):  “In the cells carrying E5 of HPV-16, there are more significant ATTS gains than losses, with a difference of about 5 significant isoforms. However, the losses outweigh the gains in the A5.”?   Also, what is meant by isoform?  Are they isoforms of E5, E6, and E7?  How do they get (or expect) up to 20 different isoforms with only one biological replicate (which I’m assuming since they didn’t mention any more in the methods)?  How do they know which isoform is “used more” (Figure 3 legend)?  These sections are not clear to me.

4.  The results for Figure 4 does mention isoforms with longer or shorter ORFs – but I’m left wondering what “used more” means – is there more mRNA for these isoforms?  Do you know if the isoform(s) that have these varying AS events and/or longer vs shorter ORFs are functional?

At this point, I am unable to continue to evaluate the merit of any additional data.  I would recommend a revision of the manuscript to provide more clarity.  I do acknowledge that my level of expertise in KEGG and GSE analyses is more limited than others, and if these comments are not in line with others reviewing this manuscript, I will happily concede to the input and advise from others.

Author Response

RESPONSE TO REVIEWERS' COMMENTS

Thank you very much for all the comments made, they really allowed us to make a better version of the manuscript. The changes suggested by you were done, as described below:

Reviewer 2 Comments

We are grateful for the suggestions that have improved our manuscript. We agree with all your observations, and we made the proper additions or modifications:

Point 1: I did not find any discussion or mention of the sample size in their research.  Did they obtain total RNA from one biological replicate of 5 x 106 cells for each of their transfected cells?  How many datasets were being analyzed?

Response 1: We discovered that the number of replicates was left out of the paper. This information is now included in the first paragraph of the section "RNA extraction and sequencing" as follows: "Total RNA was extracted in three replicates from 5 x 106 cells of each cell line using the PureLinkTM RNA Mini Kit extraction system (Ambion)."

Point 2: There is very little explanation in the methods about how they did the KEGG analysis, the parameters chosen, and what FDR <10% means. This may be appropriate for those in the field, but I feel a bit more explanation on how the analysis was performed and why they chose the genes/pathways to represent in Figure 1.  While the authors had a focus for RNA processing pathways, Figure 1 shows even higher NES scores for “Ribosome” and various autoimmune conditions.  Why are these illustrated and/or what do they mean?  There’s no discussion of these other data in Figure 1.

Response 2: I appreciate your consideration. Generally, for GSEA different ranges of FDR can be considered (from <5% to <25%) here we are listing some previously published articles for accepted FDR limits:

(A)       Tan L, Xu Q, Shi R, Zhang G. Bioinformatics analysis reveals the landscape of immune cell infiltration and immune-related pathways participating in the progression of carotid atherosclerotic plaques. Artif Cells Nanomed Biotechnol. 2021 Dec;49(1):96-107. doi: 10.1080/21691401.2021.1873798. PMID: 33480285.

(B)       Wang F, Xu X, Zhang N, Chen Z. Identification and integrated analysis of hepatocellular carcinoma-related circular RNA signature. Ann Transl Med. 2020 Mar;8(6):294. doi: 10.21037/atm.2020.03.06. PMID: 32355738; PMCID: PMC7186732.

(C)       Lee B, Mahmud I, Marchica J, DereziÅ„ski P, Qi F, Wang F, Joshi P, Valerio F, Rivera I, Patel V, Pavlovich CP, Garrett TJ, Schroth GP, Sun Y, Perera RJ. Integrated RNA and metabolite profiling of urine liquid biopsies for prostate cancer biomarker discovery. Sci Rep. 2020 Feb 28;10(1):3716. doi: 10.1038/s41598-020-60616-z. PMID: 32111915; PMCID: PMC7048821.

We focused on mRNA processing (commonly enriched in various groups) to better interpret the pathway analysis and explore deeper into these types of changes in cells. Of fact, many immune-related pathways are enriched amongst our groups and could be useful in future research.

Point 3: In the results for Figure 2, alternative splicing mechanisms are shown and the discussion mentions “gains and losses”.  Compared to what?  Each other?  Is the point that the HPV16 strains use more or less AS strategies than the HPV84 strains?  What does this mean (lines 105-107): “In the cells carrying E5 of HPV-16, there are more significant ATTS gains than losses, with a difference of about 5 significant isoforms. However, the losses outweigh the gains in the A5.”?   Also, what is meant by isoform?  Are they isoforms of E5, E6, and E7?  How do they get (or expect) up to 20 different isoforms with only one biological replicate (which I’m assuming since they didn’t mention any more in the methods)?  How do they know which isoform is “used more” (Figure 3 legend)?  These sections are not clear to me.

Response 3: “gains and losses” are compared to each other for each splicing mechanism like ATTS, A5, etc in each group. This is a kind of explaining how different mechanisms of splicing change between cells.  The “Isoform” in this paper means “different transcripts of cellular genes” and differential usage of gene transcripts between conditions is shown as “isoform switching.” 

Point 4.1: The results for Figure 4 does mention isoforms with longer or shorter ORFs – but I’m left wondering what “used more” means – is there more mRNA for these isoforms? 

Response 4.1: We include a detailed explanation about “used more” and “used less” in the first paragraph of the section "Isoform switch events and alternative splicing analysis" as follows: “Finally, the isoform(s) used more (positive dIF) is compared to the isoform(s) used less (negative dIF) to identify potential functional consequences of the isoform switch”

Point 4.2: Do you know if the isoform(s) that have these varying AS events and/or longer vs shorter ORFs are functional?

Response 4.2:

The following is mentioned in the section "Isoform switch events and alternative splicing analysis":

“To predict the consequence of each isoform switch event in coding, non-coding, and nonsense-mediated decay (NMD) sensitive categories, data from external annotation sources, including Pfam for protein domains, SignalP for signal peptides, CPAT for potential coding sequences, and IUPred2A for prediction of Intrinsically dis-ordered regions were then imported and integrated to the IsoformSwitchAnalyzeR object.”

However functional assays are a more reliable way to discover the functionality of these isoforms. 

Thank you for your time and consideration.

Luis Felipe Jave Suárez, Maryam Nasiri Aghdam,

Round 2

Reviewer 2 Report

The authors made several grammatical edits to the manuscript and added in a couple of clarifying phrases/sentences. I didn't see any substantial improvement in the interpretation of data or clarity in discussion, but as the other reviewer originally found this paper acceptable, and if the editors find that the explanation of all analyses are appropriate, I will recommend that this manuscript be considered following a careful review of the English.